# Targeted Integration of Inducible Caspase-9 in Human iPSCs Allows Efficient *in vitro* Clearance of iPSCs and iPSC-Macrophages

**DOI:** 10.3390/ijms21072481

**Published:** 2020-04-03

**Authors:** Alexandra Lipus, Ewa Janosz, Mania Ackermann, Miriam Hetzel, Julia Dahlke, Theresa Buchegger, Stephanie Wunderlich, Ulrich Martin, Toni Cathomen, Axel Schambach, Thomas Moritz, Nico Lachmann

**Affiliations:** 1RG Reprogramming and Gene Therapy, Hannover Medical School, Hannover 30625, Germany; kuhnalexis@aol.com (A.L.); janosz.ewa@mh-hannover.de (E.J.); Hetzel.Miriam@mh-hannover.de (M.H.); Moritz.Thomas@mh-hannover.de (T.M.); 2Institute of Experimental Hematology, REBIRTH, Hannover Medical School, Hannover 30625, Germany; Ackermann.Mania@mh-hannover.de (M.A.); Dahlke.Julia@mh-hannover.de (J.D.); Buchegger.Theresa@mh-hannover.de (T.B.); Schambach.Axel@mh-hannover.de (A.S.); 3RG Translational Hematology of Congenital Diseases, Hannover Medical School, Hannover 30625, Germany; 4Leibniz Research Laboratories for Biotechnology and Artificial Organs (LEBAO), Department of Cardiothoracic, Transplantation and Vascular Surgery, REBIRTH, Biomedical Research in Endstage and Obstructive Lung Disease Hannover (BREATH), Member of the German Center for Lung Research (DZL), Hannover Medical School, Hannover 30625, Germany; Wunderlich.Stephanie@mh-hannover.de (S.W.); Martin.Ulrich@mh-hannover.de (U.M.); 5Institute for Transfusion Medicine and Gene Therapy, Medical Center-University of Freiburg, Freiburg 79106, Germany; toni.cathomen@uniklinik-freiburg.de; 6Center for Chronic Immunodeficiency, Faculty of Medicine, University of Freiburg, Freiburg 79095, Germany

**Keywords:** inducible caspase 9, suicide genes, iPSC, macrophages, cell therapy

## Abstract

Induced pluripotent stem cells (iPSCs) offer great promise for the field of regenerative medicine, and iPSC-derived cells have already been applied in clinical practice. However, potential contamination of effector cells with residual pluripotent cells (e.g., teratoma-initiating cells) or effector cell-associated side effects may limit this approach. This also holds true for iPSC-derived hematopoietic cells. Given the therapeutic benefit of macrophages in different disease entities and the feasibility to derive macrophages from human iPSCs, we established human iPSCs harboring the inducible Caspase-9 (iCasp9) suicide safety switch utilizing transcription activator-like effector nuclease (TALEN)-based designer nuclease technology. Mono- or bi-allelic integration of the iCasp9 gene cassette into the AAVS1 locus showed no effect on the pluripotency of human iPSCs and did not interfere with their differentiation towards macrophages. In both, iCasp9-mono and iCasp9-bi-allelic clones, concentrations of 0.1 nM AP20187 were sufficient to induce apoptosis in more than 98% of iPSCs and their progeny—macrophages. Thus, here we provide evidence that the introduction of the iCasp9 suicide gene into the AAVS1 locus enables the effective clearance of human iPSCs and thereof derived macrophages.

## 1. Introduction

Since the development of induced pluripotent stem cells (iPSCs) by Takahashi and Yamanaka in 2006 [1], iPSCs have been proposed for innovative cell-based therapies and as a novel approach for regenerative medicine. Indeed, iPSCs and thereof derived cell types have already been successfully applied as disease modeling and drug discovery platforms [2,3,4]. However, despite immense progress in reprogramming techniques, iPSCs’ biology, and differentiation protocols, they are still not available as a standard therapy.

One reason for such cautious employment of iPSC-based cell products in patients are concerns about their pluripotent and possible tumor-initiating origin. Any residual pluripotent cells in a therapeutic product or a compromised functionality and stability of iPSC-derived cells *in vivo* may cause severe adverse effects [5]. Thus, a number of methods have been developed to limit the potential risks and move the iPSC-based therapies forward. These include strict purification of the differentiated cells, culture medium supplementation to remove iPSCs [6], use of synthetic peptides with iPSC-selective toxicity [7], or employment of suicide safety switches [8]. 

Suicide genes like the Herpes Simplex Virus thymidine kinase (HSV-TK) and the inducible Caspase 9 (iCasp9) have already been extensively studied in multiple cell types and also applied in the clinical setting [9,10,11]. However, the HSV-TK system has a number of potential drawbacks such as the dependency on actively dividing cells or potential immunogenicity of the kinase molecule. On the contrary, iCasp9 is a viable alternative to HSV-TK as it is non-immunogenic and has a significantly faster cell killing rate. iCasp9 is a synthetic monomer composed of a drug-binding domain linked to a caspase 9 pro-molecule, that upon administration of a chemical inducer of dimerization (CID) is directing cells into apoptosis within only a few hours [12,13]. In fact, the iCasp9 suicide gene strategy has been used in T cells delivered to augment immunoreactivity in patients following allogeneic hematopoietic stem cell transplantation (alloHSCT) [8]. Given these promising clinical studies, current efforts are also looking into the use of the iCasp9 as a safety switch for iPSCs and iPSC-derived cell types [14,15,16]. Moreover, recent studies demonstrated that iPSCs of various origin including murine, non-human primate and human cells, effectively undergo apoptosis upon the induction of iCasp9 by CID [14,15,17,18]. Overexpression of iCasp9 was also shown to delay iPSC-driven teratoma formation *in vivo* when CID was injected simultaneously with the iPSCs. However, teratoma growth was not affected when the CID treatment was delayed for 7 days [17]. 

Clearly direct application of non-differentiated cells does not represent the most likely scenario for iPSC-based therapeutic intervention, whereas the delivery of differentiated iPSC-progeny, such as cardiomyocytes, etc. [19,20,21] has been advocated in a number of settings. One promising cell type, gaining more attention for future use in numerous pathologic conditions are macrophages (MΦ). MΦ represent crucial players in innate immunity and are specialized to take up and process cellular debris and foreign material. Additionally, MΦ reside within organs as highly specialized tissue-resident MΦ (TRMs), supporting organ functionality and homeostasis [22]. Such TRMs include microglia in the brain, Kupffer cells in the liver or alveolar macrophages (AMΦ) in the lung and are affected in a plethora of rare diseases. As an example, impairment of AMΦ development by mutations in the GM-CSF signaling cascade causes hereditary pulmonary alveolar proteinosis (herPAP), a disease in which the absence of mature AMΦ leads to the accumulation of surfactant material in the lung. Recently, a promising approach of pulmonary MΦ transplantation (PMT) was shown to improve or even completely abrogate the herPAP disease phenotype in murine models [23,24,25]. A similar method of PMT was also described to rescue mice suffering from adenosine deaminase deficiency (ADA^-/-^) [26]. These and other data [25,27,28] strongly argue for a therapeutic role of MΦ transfer in a number of disease conditions affecting a variety of organs, and the availability of safety improved MΦ would further support the clinical translation of such concepts.

Given the option to derive functional and therapeutically effective MΦ from iPSC sources even in scalable quantities [29,30], we here aimed to develop an iCasp9 suicide approach in order to efficiently deplete iPSC and iPSC-derived MΦ (iMΦ) *in vitro*. For this purpose, we utilized transcription activator-like effector nuclease (TALEN)-based designer nucleases to introduce the iCasp9 suicide system into the adeno-associated virus integration site 1 (AAVS1) safe harbor locus of human iPSCs achieving mono- as well as bi-allelic integration of the iCasp9 gene cassette. Using these iCasp9-iPSCs, we studied *in vitro* eradication of human iPSCs and thereof differentiated MΦ following exposure to CID to establish a foundation for safety improved iMΦ.

## 2. Results

### 2.1. Generation of iCasp9 Expressing Human iPSC Lines

To generate the iCasp9-iPSCs, the transgene was inserted into the human AAVS1 site, a widely used safe harbor locus [31,32] employing TALENs as an efficient tool for genome editing [33]. An AAVS1-specific TALEN pair previously shown to generate low cytotoxicity [34] was used to target the *PPP1R12C* gene within the AAVS1 locus and insert a transgene cassette containing the iCasp9 gene under control of the hybrid CMV early enhancer/chicken β-actin (CAG) promoter. The transgene cassette also contains a puromycin selection gene and a splice acceptor followed by a self-cleaving peptide 2A. After targeted integration, the selection marker will be expressed from the endogenous *PPP1R12C* promoter to allow puromycin-based selection. To direct the template to the target locus, right and left homology arms spanning ~750 bp at the respective end of the construct were used to allow for homologous recombination within the AAVS1 locus (Figure 1A). 

An iPSC line derived from CD34-positive hematopoietic cells of a healthy donor from which the reprogramming cassette was previously excised (CD34iPSC16exi) was employed and nucleofected with the above described TALENs and the donor template. After puromycin selection, single colonies were transferred to irradiated murine embryonic fibroblast (MEF) feeder cell-seeded culture plates and expanded (Figure 1B). Analysis of the genomic integration of the transgene was performed by a PCR-based genotyping using primers binding at the 5′ (p1+p3) or 3′ end (p2+p4) of the integrated cassette to allow for the identification of targeted integration. Not edited AAVS1 alleles were determined using the primer pair p1+p2. This screen enabled us to identify 18 clones (53%) with mono-allelic and 16 clones (47%) with bi-allelic targeted integration amongst a total of 36 clones analyzed (Figure 1C).

### 2.2. Effective Killing of iCasp9-iPSCs

One clone each with mono- (clone 12) and bi-allelic (clone 10) transgene integration (iCasp9-mono, iCasp9-bi respectively) was chosen for further experiments. The pluripotency status of iCasp9-mono, iCasp9-bi, and the control CD34iPSC16exi cells was confirmed by positive alkaline phosphatase staining and PCR-based expression analysis of *OCT4, NANOG*, and *SOX2* showing levels comparable with the H9 human embryonic stem cell (ESC) line (Appendix A). The iCasp9 dimerizing drug AP20187 was applied to iPSCs cultured on MEF feeder cells at concentrations ranging from 0.01 nM to 10 nM. To determine the efficacy of iCasp9 induced cell death, 24 h after AP20187 supplementation cells were stained with propidium iodide (PI) to visualize apoptotic cells (Appendix A). At standard cell culture condition, i.e., without AP20187 added, a background level of ~20% PI^+^ cells was detected and similar levels of PI^+^ cells were observed in the control CD34iPSC16exi clone at all concentrations of the CID (Figure 2A). In contrast, iCasp9-mono and iCasp9-bi cells displayed a PI^+^ signal of 60%-95% already at 0.01 nM- the lowest concentration of the CID applied in this setting (Figure 2B,C). At higher concentrations of AP20187, the percentage of PI^+^ iCasp9-mono and iCasp9-bi cells increased even further to levels of 80%-98%. The effect of the iCasp9 dimerizer on both iCasp9-iPSCs clones was also visualized by bright field microscopy displaying distinct morphological changes, including loss of colony unity and cell adherence (Figure 2D).

To look closer into the kinetics of iCasp9-mediated killing, iCasp9-bi cells were treated with 1 nM AP20187 and the percentage of viable (PI^-^) cells was analyzed over a period of 24 h. Marked reduction in cell viability was already observed after 1 h and continued thereafter reaching 25% and 10% of viable cells when compared to the starting value after five and 24 h, respectively (Figure 2E, Appendix A). 

As the MEF feeder cells used in our initial set of experiments are not sensitive to AP20187 and thus may have affected the data, the experiments were repeated in a feeder-free setup, where the iPSCs were cultured on matrigel. Similarly to the MEF-based cultures, again background levels of 15%-25% of PI^+^ cells were observed (Figure 2F). Killing efficiency of >95% and >98% was observed in both, iCasp9-mono and iCasp9-bi iPSCs for AP20187 concentrations of 0.01 nM and 0.1 nM respectively (Figure 2G,H).

To analyze the fate of potential “break-through” surviving cells following CID treatment more stringently, secondary cultures of cells exposed to 1 nM of CID were initiated. Whereas in all control experiments iPSCs regrew to confluency, no colony growth was detected in the iCasp9 iPSCs treated with 1 nM of the dimerizer up to 14 days after reseeding (Appendix A).

Thus our results demonstrate that supplementation with 0.1-10 nM CID is highly effective in killing >98% of iCasp9-iPSCs and the remaining cells are not able to regrow cell colonies after iCasp9 induction. Moreover, in our hands, this effect could be obtained to a similar extent with a mono- or bi-allelic iCasp9 integration.

### 2.3. Macrophage Differentiation from iCasp9-iPSCs

To evaluate the efficacy of the iCasp9 suicide strategy in the hematopoietic progeny of iPSCs, the iCasp9-iPSCs were differentiated into iMΦ. The differentiation was performed using our well-established protocol allowing for weekly harvests of iMΦ over prolonged time periods [30]. In the current study, no adverse effect of the iCasp9 integration cassette on the efficacy of cellular differentiation was observed when CD34iPSC16exi and iCasp9-mono or iCasp9-bi cells were compared. Nor did expression of the iCasp9 affect the iMΦ morphology when iMΦ derived from the parental CD34iPSC16exi or the iCasp9-iPSC clones were compared on cytospin staining (Figure 3A). In addition, iMΦ stained positive for standard myeloid and MΦ surface markers CD45, CD11b, CD14, and CD163 (Figure 3B). To assess whether the expression of the iCasp9 transgene might affect the functionality of iMΦ, the secretion of IL-6 was evaluated. Similar levels of IL-6 were secreted by iMΦ derived from control CD34iPSC16exi and iCasp9-iPSCs, (Figure 3C) arguing against a profound effect of iCasp9 expression on iMΦ functionality. Of note, no secretion of human IL-6 could be detected in mouse embryonic fibroblast (MEFs), while MΦ generated from peripheral blood (PB-MΦ) showed reduced secretion of IL-6 compared to iMΦ, an observation already made in previous studies [30]. 

### 2.4. Induction of Killing in iMΦ

To evaluate whether iCasp9 is equally effective in iMΦ as in the parental iPSCs, iMΦ derived from iCasp9-bi cells were incubated with AP20187 for 24 h. In the absence of CID, parental CD34iPSC16exi and iCasp9-bi cells showed similar background level of dead cells at the 24 h time point based on FSC/SSC gating (Appendix A), arguing against specific toxicity of iCasp9 expression in iMΦ. In contrast to CD34iPSC16exi-derived iMΦ, however, a CID concentration as low as 0.01 nM already induced cell death in >90% of iCasp9-bi-derived iMΦ (Figure 4A,B). Treatment with higher concentrations of CID increased the number of dead cells even further to >98% (Figure 4B). These findings were confirmed by microscopic images of the iMΦ cultures 24 h after CID exposure (Figure 4C). While there were no morphological changes detectable in cultures of CD34iPSC16exi-derived iMΦ in either untreated or dimerizer treated conditions, the iCasp9-bi iMΦ lost adherence, clumped together and clearly displayed an apoptotic phenotype following iCasp9 activation. Thus, these data clearly show the efficacy of the iCasp9 suicide system also in iPSC-derived hematopoietic cells, such as iMΦ, following insertion of a suitable iCasp9 expression cassette into the AAVS1 safe harbor locus.

## 3. Discussion

Clinical translation of the iPSC-based therapies remains a high priority in cell therapy and regenerative medicine research, however, the potential contamination with remaining pluripotent cells and unwanted side effects of transplanted cells are limiting this approach. Importantly, the clinical translation of iPSC-derived mature effector cells is already taking place. While the first trial utilizing iPSC-derived retinal cells was performed in 2014 (UMIN000011929) [35], current efforts are underway using either iPSC-derived mesenchymal stem cells to treat steroid-resistant acute graft-versus-host disease (NCT02923375), iPSC-derived dopaminergic neurons as a brain graft for Parkinson’s disease (UMIN000033564) or iPSC-derived cardiomyocyte sheets as a remedy for ischemic hearts [36]. Despite these promising endeavors, the debate still remains as to whether these therapies would pose any potential risks for patients. Therefore, additional safety measures could advance the development of iPSC-based treatments for different disease indications.

Based on this discussion and the therapeutic activity of macrophages in a variety of disease settings, we have investigated the efficacy and safety of the iCasp9 system in iPSC and thereof derived macrophages. We demonstrated the effective elimination of both cell types from *in vitro* cell culture systems upon application of dimerizer with no differences in morphology or viability of iCasp9-expressing or non-expressing cells even in the absence of CID. To this end, we established a successful targeted integration of the iCasp9 construct into the AAVS1 locus using TALE-nuclease technology. To express the iCasp9 transgene cassette, we used the CAG promoter, which in our hands allowed for effective elimination of >98% of genetically modified iPSCs and thereof derived macrophages within 24 h of dimerizer treatment. Cell death was induced at CID concentration as low as 0.1 nM. This compares to previous studies using an EF1α promoter-driven iCasp9 transgene cassette, which was delivered by lentiviral transduction into murine iPSCs describing a killing efficiency of >90% for iPSC [17]. Likewise, the expression of iCasp9 from an EF1α promoter was previously reported for human iPSCs with >95% killing efficiency. Similar rates of cell death were observed in iPSC-derived mesenchymal stromal cells or neurons [14,15]. However, in the aforementioned studies, maximal induction of cell death required 10 nM of CID, which is 100 times more than in our study.

A potential explanation for the higher sensitivity of the iCasp9 system to the CID in our hands might be the targeted integration of the transgene cassette into the AAVS1 locus and a strong CAG promoter-driven expression of the iCasp9. It has been shown that iPSCs with random integration of iCasp9 might downregulate the iCasp9 level and a dimerizer-resistant cell population may arise [17,18]. Furthermore, integration of the iCasp9 transgene into the safe harbor locus may prevent the promoter region from epigenetic silencing [32,37]. Of note, effective transgene expression from cassettes integrated into the AAVS1 safe harbor site was also demonstrated in other studies using the same or similar TALEN technology as we did [34,38,39].

While we effectively induced apoptosis in iPSCs and their hematopoietic progeny, we could not ablate 100% of the cells, with a portion of approximately 2% PI negative cells remaining in the cultures. Importantly, however, this small population of remaining iCasp9-iPSCs in contrast to CD34iPSC16exi control cells was not able to form new iPSC colonies following CID exposure when seeded in secondary cultures.

As mentioned before, here we present an iCasp9 safety switch approach, which is based on a constitutive expression of the transgene in both the iPSCs and their derivatives. In principle, this strategy allows for a complete elimination of the graft in an *in vivo* cell therapy setting. However, in certain situations, only partial elimination of donor cells such as remaining pluripotent cells versus adequately differentiated mature cells may be warranted. This can be achieved by either specific targeting of dividing cells as a potential tumorigenic source [40] or cell type-specific promoters, which selectively drive the expression of the iCasp9 transgene either in pluripotent cells or the differentiated cell types [41].

Our studies establish a proof of concept for the efficient killing of iMΦ expressing the iCasp9 gene. This bears considerable importance as MΦ-based cell therapies are currently being evaluated in a large number of disease settings [42,43]. Importantly, also iMΦ have already shown considerable therapeutic activity in pre-clinical studies. This includes the targeting of human gastric cancer cells (NUGC-4) with genetically engineered iMΦ overexpressing IFN-β in a murine intraperitoneal tumor model [28]. Another very promising therapeutic application of MΦs is direct transplantation into the lungs to treat herPAP [25], bacterial infections [29], or the lung associated phenotype of adenosine deaminase (ADA) deficiency [26]. 

In the aforementioned therapeutic scenarios, the *in vivo* administration of the CID may be performed directly into the lung or via the intravenous route. Notably, intravenous and intraperitoneal delivery of the dimerizer has been well studied and described to effectively induce iCasp9-mediated cell death in teratomas and tumors thereby limiting their growth in murine models [15,17,44]. Even more important, intravenous CID administration was demonstrated to deplete T cells and resolve graft versus host disease symptoms in patients treated with donor lymphocyte infusions following alloHSCT within 2 h after CID administration [8]. Based on these data, it can be speculated that depletion of iMΦ from hematopoietic organs and the systemic circulation following the CID application may result in similar kinetics. However, kinetics of iCasp9-induced cell death in alveolar macrophages or TRM populations residing in other organs have not yet been studied. Therefore, further tissue penetration studies for the CID seem necessary. Previous safety studies have already established that the iCasp9 dimerizer is well tolerated in humans even at high doses of 1 mg/kg (resulting in serum levels >800 nM) [45]. This would probably suffice to reach the CID concentration required to eliminate the iMΦ from the lung or other organs in case of adverse effects. Nevertheless, as a next step towards clinical applicability of our AAVS1-targeted iCasp9 expression strategy vigorous *in vivo* studies proving efficacy of the concept are mandatory.

In summary, our studies prove that the introduction of the iCasp9 suicide gene expression cassette into the AAVS1 locus under control of the hybrid CAG promoter enables the effective clearance not only of human iPSCs but also iMΦ. The data establishes an important fail-safe mechanism in the context of pluripotent cells and thereof derived effector cells and thus may further facilitate the translation of iMΦ and other iPSC-derived products for future use in cell therapy applications.

## 4. Materials and Methods 

### 4.1. Plasmids and Donors

The AAVS1 locus-specific TALEN expression plasmids were previously described by Mussolino et al. [34]. The AAVS1.iCaspase9 donor was constructed using standard cloning technologies containing homology arms of ~750 bp flanking the puromycin resistance gene, a splice acceptor site (SA), a self-cleaving peptide sequence (2A), a BGH poly(A), the CAG promoter driving the expression of the iCasp9 cDNA followed by the HSV-TK Poly(A).

### 4.2. Cell Culture

The reprogramming cassette harboring the “Yamanaka factors” Oct4, Sox2, Klf4, and c-Myc was excised from the healthy human CD34iPSC16 iPSC line (previously described in [46]) resulting in the CD34iPSC16exi line used in this study. iPSCs were maintained on CF-1 MEF feeder cells in Knock Out Dulbecco’s modified Eagle medium (KO-DMEM) supplemented with 20% knock out serum replacement, 1% penicillin/streptomycin, 0.1 mM non-essential amino acids, 1 mM L-glutamine, 0.1 mM ß-mercaptoethanol and 10 ng/mL basic fibroblast growth factor (bFGF) (Peprotech, Hamburg, Germany) at 37 °C with 5% CO_2_ incubation. Passaging of the iPSCs on new MEF cells was performed every seven to ten days using 2 mg/mL collagenase IV (Invitrogen, Carlsbad, CA, USA). MEF cells were seeded one day prior passaging of iPSCs in Dulbecco´s modified Eagle medium (DMEM) with 1% penicillin/streptomycin, 0.1 mM non-essential amino acids, 1 mM L-glutamine and 0.1 mM ß-mercaptoethanol. All cell lines were cultured at standard conditions at 37°C and 5% CO_2_. The reagents were obtained from Thermo Fisher Scientific, MA, USA.

### 4.3. Nucleofection and Gene Editing of Human iPSC

Nucleofection of human CD34iPSC16exi iPSCs was performed using the Nucleofector 2b device (Lonza, Basel, Switzerland) and the Human Stem Cell kit 2 (Lonza, Basel, Switzerland) according to the manufacturer´s instructions. In short, iPSCs were grown to sub-confluency and disaggregated into single cells using TrypLE (Thermo Fisher Scientific, MA, USA). For nucleofection, 2x10^6^ cells were resuspended in nucleofection solution supplemented with 3.8 µg donor DNA plasmid or 1.2 µg pMax-GFP transfection control and 0.6 µg each TALEN DNA plasmid. Transfection efficiency was determined by flow cytometry 48 h post nucleofection and was in the range of 75%. Correctly targeted cells were selected in standard medium supplemented with 0.5 µg/mL puromycin for seven to ten days, starting 48 h post nucleofection. Positive colonies were picked, further expanded and initially analyzed by PCR to identify targeted clones.

### 4.4. Genotyping of Gene-Edited iPSCs 

The extraction of genomic DNA was performed with the GenElute mammalian genomic DNA miniprep kit (Sigma Aldrich, MO, USA). For targeted integration PCRs Phire Hot Start II Polymerase (Thermo Fisher Scientific, MA, USA) was used under standard conditions. To detect the 5‘-junction and 3‘-junction of targeted integrated (TI) donor DNA and the genome, the following primers were used: p1-5‘-TI_AAVS1_fwd: 5‘-CCAGCTCCCATAGCTCAGTCTG-3‘, p3-5‘-TI_Puro_rev: 5‘-GGTCCTTCGGGCACCTCGAC-3‘, p2-3‘-TI_AAVS1_rev: 5‘-GGGCTCAGTCTGAAGAGCAGAG-3‘, p4-3‘-TI_ iCaspase9_fwd: 5‘-TCTAGTTTGCCCACACCCAG-3‘. To discriminate between clones that were targeted mono-allelic or bi-allelic, the p1-TI_AAVS1_fwd and p2-TI_AAVS1_rev primers were used. 

### 4.5. Real-Time Quantitative Reverse Transcription PCR Analysis

For detection of gene expression levels, cells were cultured under standard conditions, harvested and RNA was isolated using phenol-chloroform followed by cDNA synthesis using the RevertAid H Minus First Strand cDNA Synthesis Kit (Thermo Fisher Scientific, MA, USA). Expression of pluripotency related factors was assessed using TaqMan Universal Master Mix II (with UNG) (Thermo Fisher Scientific, MA, USA) using primers for human *POU5F1* (OCT4), human *SOX2,* human *NANOG* and human *GAPDH* (Thermo Fisher Scientific, MA, USA). For all quantitative PCRs, the StepOne Real Time PCR System from Applied Biosystems (Thermo Fisher Scientific, MA, USA) was used.

### 4.6. Alkaline Phosphatase Staining 

For visualization of the alkaline phosphatase activity, the Stemgent Alkaline Phosphatase (AP) Staining Kit (Pelo Biotech, Planegg, Germany) was used and applied according to the manufacturer´s instructions. In brief, iPSCs were cultured under standard conditions to 40% confluency followed by fixation and AP staining. Pictures were taken within the next one to two days.

### 4.7. Hematopoietic Differentiation of Human iPSCs Toward Monocytes/Macrophages

A previously described an embryoid body (EB)-based hematopoietic differentiation protocol was used for the generation of monocytes/macrophages [30]. In short, iPSCs were cultivated in standard medium supplemented with 10 ng/mL bFGF. Prior to the differentiation step, the bFGF was excluded from the medium. After disruption of the iPSC colonies using collagenase IV (Invitrogen, Carlsbad, CA, USA), they were transferred to a 6-well suspension plate in standard medium containing 10 µM Rock Inhibitor (Y-27632, Tocris Bioscience, Bristol, UK) on an orbital shaker (100 rpm) to generate EBs. After five days, EBs were manually transferred onto a 6-well tissue culture plate in differentiation medium I (X-Vivo 15 (Lonza, Basel, Switzerland) supplemented with 1% penicillin/streptomycin, 1 mM L-glutamine, 0.1 mM ß-mercaptoethanol, 25 ng/mL human IL-3 and 50 ng/mL human M-CSF). EBs were allowed to attach to the plate to form myeloid-cell forming complexes (MCFCs) over a period of seven days. The production of monocytes/macrophages from the MCFCs occurred from day ten to day 15 onwards. IMΦ were harvested from the supernatant once a week, filtered through a 100 µm mesh and further differentiated for five to ten days on tissue culture plates in differentiation medium II (RPMI1640 medium supplemented with 10% fetal calf serum, 1% penicillin/streptomycin, 1 mM L-glutamine, and 50 ng/mL human M-CSF).

### 4.8. Flow Cytometric Analysis

The expression of different surface markers was analyzed using the following antibodies: hCD45-PE, hCD14-PE, hCD163-APC, hCD11b-APC, and isotype controls mouse-IgG1κ-PE or APC, all from eBioscience, San Diego, CA, USA. Expression was analyzed using a FACS Calibur (Becton & Dickinson, Franklin Lakes, NJ, USA) or CytoFLEX S (Beckman Coulter, CA, USA). 

### 4.9. Cytospins

Cytospins were performed with 2x10^4^ monocytes/macrophages with a Shandon cytocentrifuge (Thermo Scientific, Langenselbold, Germany). Following May–Grünwald–Giemsa staining, cells were allowed to dry and were covered before evaluation by bright field microscopy.

### 4.10. IL-6 Secretion Analysis

For human IL-6 secretion analysis, iPSC-derived monocytes/macrophages were terminally differentiated in 50 ng/mL human M-CSF for 3-7 days. As a positive control, macrophages from peripheral blood mononuclear cells (PBMCs) were used while mouse embryonic fibroblasts were used as a negative control. PBMCs were isolated from peripheral blood of healthy volunteers using gradient centrifugation with Biocoll Separating Solution (Merck, Germany) for 40 min, 400 g. The healthy donors gave written informed consent according to the local ethical committee at Hannover Medical School. PBMCs were differentiated in RPMI1640 medium with 10% fetal calf serum, 2 mM L-glutamine, 1% penicillin-streptomycin (all Invitrogen, CA, USA) supplemented with 10 ng/mL hM-CSF and 10 ng/mL hIL-3 (Peprotech, Hamburg, Germany) for first 7 days and only 10 ng/mL hM-CSF for next 7 days. IMΦ, murine embryonic fibroblasts and PBMC-derived MΦ were cultivated in 96-well tissue culture plates for 24 h at a density of 6x10^4^ cells/well. After starvation for 24 h in X-Vivo 15 medium, iMΦ were either left unstimulated or were stimulated with 1 µg/mL LPS for another 24 h. Supernatants were collected and analyzed using the human IL-6 human uncoated ELISA Kit (Thermo Fisher, Vienna, Austria) according to the manufacturer´s instructions.

### 4.11. Induction of Apoptosis with the Chemical Inducer of Dimerization AP20187

1-2x10^5^ iPSCs were seeded on MEF- or Matrigel-coated 12-well tissue culture plates and grown to sub-confluency before the functionality of the suicide gene was assessed by adding the CID (AP20187, B/B Homodimerizer, Clontech Laboratories, CA, USA) for 24 h with concentrations ranging from 0 nM- 10 nM. Twenty-four hours after supplementation with CID the cells were reseeded onto a 6-well tissue culture plate, kept for 14 days and an appearance of iPSC colonies was observed. 

Monocytes/macrophages were terminally differentiated in differentiation medium II for 5 days and seeded in 96-well tissue culture plates at a density of approximately 6-10x10^4^ cells/well before being exposed to 0 nM- 10 nM AP20187 for 24 h. For flow cytometric analysis, cells were dissociated into single cells using TrypLE and stained with PI (according to the manufacturer´s instruction). Additionally, pictures of the cells in culture plates were taken at the indicated time points.

### 4.12. Statistical Analysis

For all statistical analyses One-Way ANOVA, Dunnett’s post hoc test was performed using Graphpad Prism 6 (GraphPad Software, CA, USA).

## Figures and Tables

**Figure 1 ijms-21-02481-f001:**
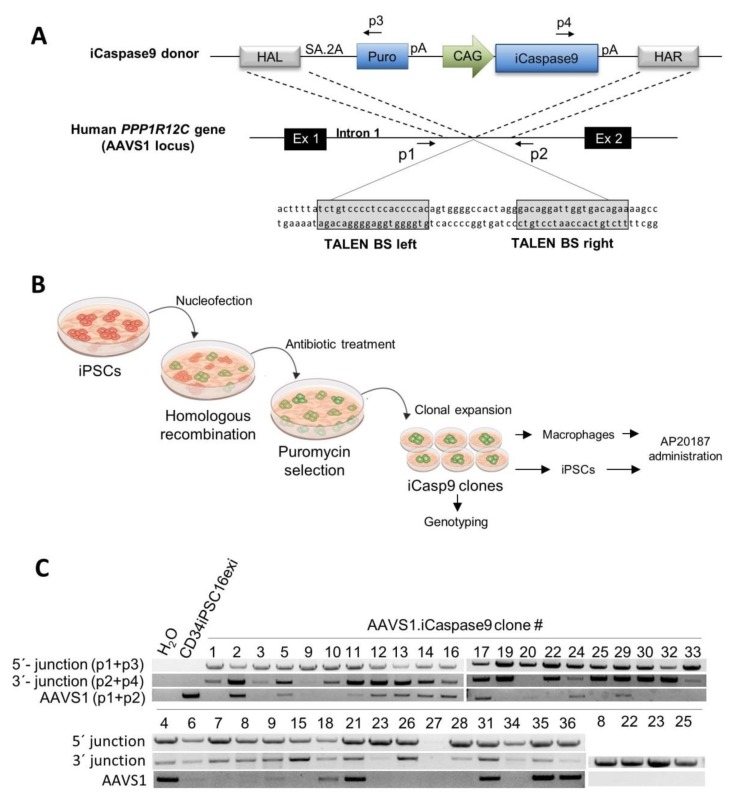
Generation and genotyping of adeno-associated virus integration site 1 (AAVS1)-targeted human induced pluripotent stem cells (iPSCs). (**A**) Scheme of the AAVS1 target site and the iCasp9 donor. The donor template targets intron 1 between exon 1 and exon 2 of the AAVS1 locus. The inducible Caspase 9 (iCasp9) is expressed from the CMV early enhancer/chicken ß-actin promoter (CAG) promoter. Arrows indicate the primers (p1-p4) that are used for genotyping. (**B**) Experimental set-up of the generation and processing of iPSC clones targeted with the iCasp9 transgene within the AAVS1 site. (**C**) PCR-based verification of the site-specific integration of the donor cassette in 36 puromycin-selected iPSC clones. HAL—homology arm left; HAR—homology arm right; SA—splice acceptor; 2A-–self-cleaving peptide; Puro—puromycin resistance gene; CAG—CMV early enhancer/chicken ß-actin promoter; pA—poly-A site; BS—binding site.

**Figure 2 ijms-21-02481-f002:**
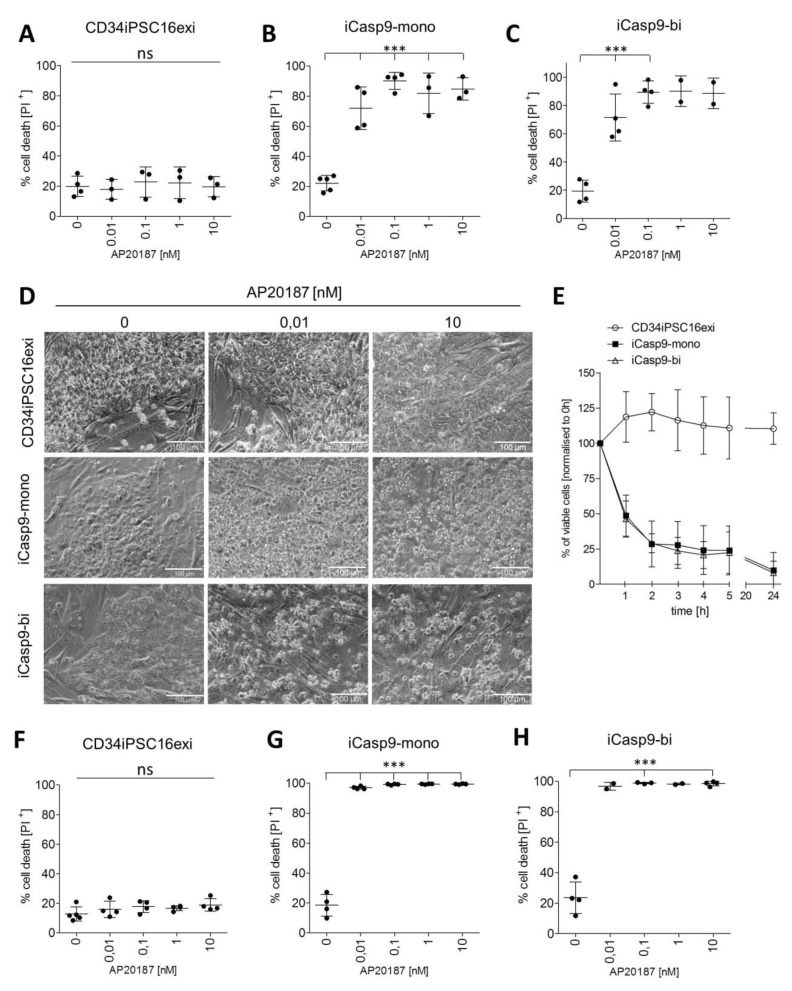
Induction of the iCasp9-mediated apoptosis in iPSCs. Percentage of dead cells according to propidium iodide (PI) staining after administration of 0.01 to 10 nM of the AP20187 dimerizer in CD34iPSC16exi iPSCs (**A**), iCasp9-mono iPSCs (**B**), and iCasp9-bi iPSCs (**C**) cultured on MEF feeders. (mean ± SD, ***P < 0.0001, ns—not significant) (**D**) Phase-contrast images of CD34iPSC16exi as well as iCasp-mono, and -bi iPSCs 24 h after adding the iCasp9 inducer. (scale bar—100 µm) (**E**) Supplementation of CD34iPSCl16exi iCasp9-mono and iCasp9-bi (clone #1) iPSCs with 1 nM of AP20187 over a period of 24 h. (n = 3) (**F**,**G**,**H**) Percentage of dead cells according to propidium iodide (PI) staining after administration of 0.01 to 10 nM of the AP20187 dimerizer in CD34iPSC16exi iPSCs (F), iCasp9-mono iPSCs (G), and iCasp9-bi iPSCs (H) cultured on Matrigel. (mean ± SD, ***P < 0.0001, ns—non significant).

**Figure 3 ijms-21-02481-f003:**
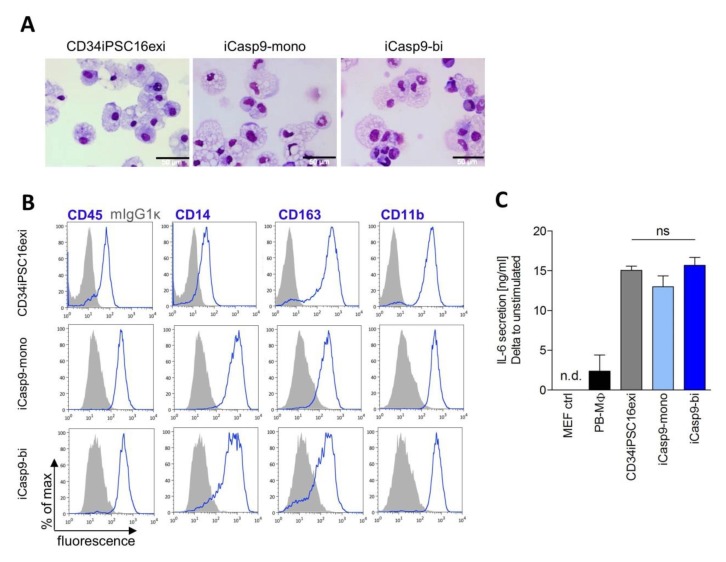
Characterization of the iPSC-derived MΦ. (**A**) Morphology of CD34iPSC16exi-iMΦ and iCasp9—iMΦ on May-Grünwald-Giemsa-stained cytospins (scale bar—50 µm). (**B**) Flow cytometric detection of hematopoietic (CD45) and MΦ-specific (CD14, CD163, CD11b) surface marker expression on iMΦ. (**C**) Secretion of human IL-6 upon stimulation with lipopolysaccharide (LPS). iMΦ from human iPSC lines CD34iPSC16exi (non-targeted), iCasp9-mono (clone #12) or iCasp9-bi (clone #1) were used. MEF cells served as a negative control and human PB-MΦ as a positive control (n = 2-3). n.d, not detected (<1.6 pg/mL); ns, not significant.

**Figure 4 ijms-21-02481-f004:**
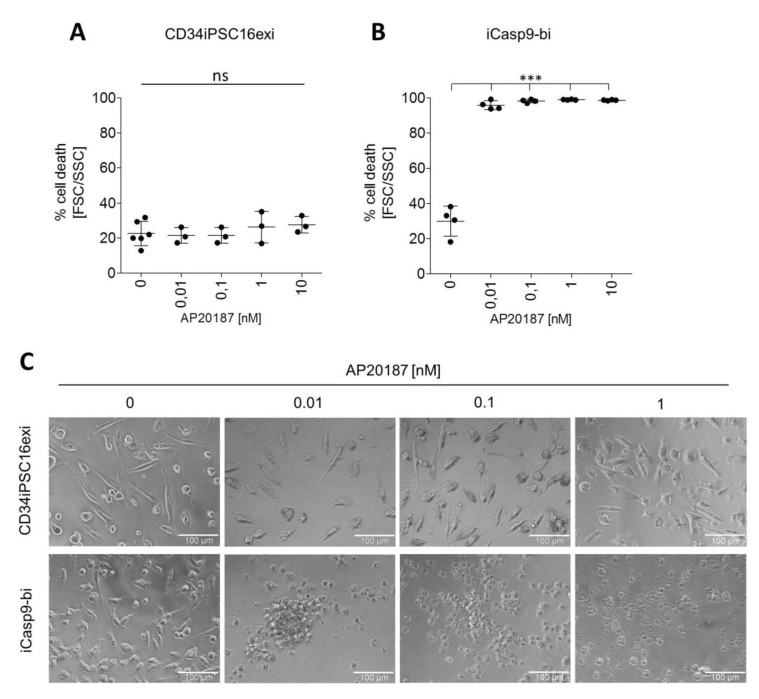
Induced killing of the iCasp9-expressing iMΦ. Percentage of dead cells according to FSC/SSC gating 24 h after application of 0 to 10 nm AP20187 in CD34iPSC16exi iMΦ (**A**) and iCasp9-bi iMΦ (**B**). (mean ± SD, ***P <0.0001, ns—non significant) (**C**) Phase-contrast imaging of the iMΦ 24 h after treatment with different concentrations of AP20187. (scale bar—100 µm).

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
