# Peer review of "Targeted Integration of Inducible Caspase-9 in Human iPSCs Allows Efficient in vitro Clearance of iPSCs and iPSC-Macrophages"

_ijms, 2020, doi:10.3390/ijms21072481_

Round 1

Reviewer 1 Report

The authors have addressed the concerns, and I have no more comments on the manuscript.

Author Response

We are pleased to read that Reviewer agrees with our revised version of the manuscript.

Reviewer 2 Report

The manuscript has been corrected according to the comments.

Author Response

(The authors gave the same response as above.)

Reviewer 3 Report

The authors have addressed most of my minor concerns. However, they chose not to pursue experiments to demonstrate clearance of iPSC or iPSC-derived macrophages, which I consider to be a critical experiment. Without evidence of physiological clearance, the title needs to be incorporate "in vitro" to accurately reflect the content of the manuscript.

Author Response

We appreciate the comment of Reviewer#3 on changing the title of the manuscript. To emphasize the in vitro focus of the manuscript, the title was modified accordingly. In addition and according to the suggestion of Reviewer#3, we also adapted the introduction paragraph.

This manuscript is a resubmission of an earlier submission. The following is a list of the peer review reports and author responses from that submission.

Round 1

Reviewer 1 Report

The study optimizes existing approaches to the clearance of human iPSCs and macrophages derived from them.

The authors do not take into account the possibility of the effect of Caspase-9 (iCasp9) on the viability of macrophages after 24 hours. It is worth introducing this information into discussions.

Reviewer 2 Report

Dear Editor,

Thank you for giving me this opportunity to review this manuscript. In this study, the authors successfully integrated inducible casp9 in hiPSCs, which remains the pluripotency and can be differentiated into functional iPSC-macrophages. The purpose of developing such apoptosis-inducible hiPSCs and hiPSC-macrophages is to deplete these cells in a controllable manner. In this study, the authors treated cells with an iCasp9 dimerizing drug (AP20187) to induce cell apoptosis in vitro. However, since iPSCs are given in animals or patients, it will be of clinical value if such apoptosis can be induced in vivo. Therefore, it’s critical to show how the iCasp9 dimerizing drug can be delivered to the target cells in vivo and whether it is effective in vivo.

Reviewer 3 Report

In this manuscript, Lipus et al. integrated an inducible caspase-9 (iCasp9) into human iPSCs, a suicide program to serve as a safeguard clearance mechanism to tackle potential hazardous effects of iPSC therapy. They showed that iCasp9-iPSCs, and their macrophage progenies, can be effectively cleared in vitro.

The experimental design is neat and simple. There are no major scientific flaws. However, the study is redundant with a large number of previous publications on iPSC clearance mechanisms. More importantly, it is very puzzling to see a need for iPSC-derived macrophages, especially because macrophages can be easily derived from patient PBMCs, without the need to derived using a dramatically more expensive iPSC approach. Not to mention the controversy on macrophage transplantation therapy.

It is crucial to show elimination of iPSC or their progenies in vivo. Also, it is also hard to fathom what an “effective” clearance mechanism would be. Granted that you could achieve 99.99% apoptosis, but the remaining 0.01% could be sufficient to initiate tumors or other negative events.

There are also a few technical concerns:

Figure 3C misses a negative and a positive control.

Quite a few data points in a couple of figures only have n=2, making it impossible to do statistical analyses. One may even wonder whether the data could be reproduced for a third time. Figure 2E only has n=1?

Figure legends were written in a way as if they better fit the results sections.